# The Effect of Photoperiodic Conditions on GnRH/LH Secretion in Ewes

**DOI:** 10.3390/ani12030283

**Published:** 2022-01-24

**Authors:** Kamila Kopycińska, Karolina Wojtulewicz, Andrzej Przemysław Herman, Dorota Tomaszewska-Zaremba

**Affiliations:** The Kielanowski Institute of Animal Physiology and Nutrition, Polish Academy of Sciences, Instytucka 3, 05-110 Jabłonna, Poland; kopycinskakamila@gmail.com (K.K.); a.herman@ifzz.pl (A.P.H.)

**Keywords:** ewes, reproduction, diurnal changes, LH, GnRH, melatonin

## Abstract

**Simple Summary:**

During the course of evolution, animals have evolved biological rhythms that are associated with changes in the lighting and temperature of their environment. Females in most breeds of sheep are seasonal breeders, with ovulatory cycles occurring in the autumn and winter and anovulation in the spring and summer. Secretion of gonadotropin releasing hormone and luteinizing hormone, the main hormones regulating reproduction in females, displays a circadian pattern; however, data concerning the day/night differences in their secretion in ewes are incomplete. The aim of the undertaken study was to determine the day/night differences in the secretion of gonadotropin releasing hormone and luteinizing hormone in follicular phase and anestrous ewes. It was demonstrated that secretion of investigated hormones is subject to diurnal and seasonal changes. The observed reduction in luteinizing hormone release, a few hours after the sunset, seems to be universal for both the anestrus and follicular phase. It could be concluded that the nocturnal suppression of luteinizing hormone secretion in follicular phase ewes may be a mechanism moving this hormone surge to the early morning.

**Abstract:**

Secretion of gonadotropin releasing hormone (GnRH) and luteinizing hormone (LH) displays a circadian pattern. Data concerning differences in daily GnRH/LH secretion during different seasons in sheep are fragmentary. The aim of the study was to determine day/night differences in GnRH/LH secretion in the follicular phase and in the anestrous ewes. The studies were performed on Blackhead ewes (*n* = 24). Ewes from each season were divided into two groups of six animals (day and night group). The animals were euthanized 5 h after sunset or 5 h after sunrise and blood was taken to determine LH and melatonin concentrations. In the hypothalamus, the expression of GnRH and gonadotropin releasing hormone receptor (GnRHR) was determined. In the anterior pituitary, the expression of mRNA encoding subunit β of LH (LHβ) and GnRHR was assayed. Our study showed that GnRH/LH secretion is subject to diurnal and seasonal changes. The observed reduction in LH release, a few hours after the sunset, seems to be universal for both the anestrus and follicular phase, when the processes occurring at the hypothalamus are more equivocal. It could be concluded that the nocturnal suppression of LH secretion in follicular phase ewes may be a mechanism moving the LH surge to the early morning.

## 1. Introduction

As a result of evolution, animals have developed biological rhythms, thanks to which physiological processes run in a rhythmic manner. These rhythmic changes are related to changes in lighting and ambient temperature [1]. Sheep are seasonal breeders, with the breeding season occurring in the fall-winter period and characterized with ovulation cycles, and the anestrous season in spring and summer with a lack of ovulation [2]. Seasonal reproduction is the consequence of profound changes in reproductive neuroendocrine activity. In vertebrates, the regulation of reproductive activity is maintained by the hypothalamic-pituitary-gonadal axis (HPG). The main neurohormone regulating the processes of reproduction is gonadotropin releasing hormone (GnRH)—a neuropeptide released in a pulsatile manner to ensure maintenance of luteinizing hormone (LH) and follicle stimulating hormone (FSH) secretion from the pituitary. In ewes, GnRH is synthesized in neuros, which perikarya are, among others, in the preoptic area (POA), and a majority of these neurons send projections to the median eminence (ME), where GnRH is released [3]. In ewes, there is evidence that the seasonal onset of the estrous cycle is the consequence of alterations in the negative feedback effect of estradiol (E2) [4].

The main external signal that regulates changes in seasonal GnRH secretion is changing day length (photoperiod). Photoperiodic information is transferred to the reproductive system through a retinohypothalamic pathway—the photoperiodic input is received through eyes, then transmitted via a multi-synaptic pathway to the pineal gland, which transduces the photic signal into a chemical—melatonin secretion [4]. It was demonstrated that melatonin with blood reaches the peripheral tissues and is delivered by the cerebrospinal fluid (CSF) to the brain [5]. Thiéry et al. [6] showed that the turnover rate of CSF in ewes changed according to the light-dark cycle, so this could affect the melatoninergic message to the suitable sites in the brain. Melatonin acts on GnRH pulse activity at the premammilary hypothalamus (PMH) via melatonin receptor type 1 (MT1) [4] and very complex neural network [7]. In sheep, complex endocrine processes involving different neuromediator systems such as catecholaminergic, kisspeptin, gamma-aminobutyric acid (GABA), prostaglandins, opioids, neuropeptide Y (NPY) or corticotropin-releasing hormone (CRH), and neuronal plasticity have been shown to play a role in these processes [4]. In ewes, regulation of GnRH depends on the physiological status of the animal and is different during the breeding season and anestrus. It was demonstrated that the functions of β-endorphinergic, catecholaminergic, and GABA-ergic neuronal systems in the control of GnRH release change during seasonal anestrus and in the course of the estrous cycle [8].

The literature data suggest that most of the physiological processes, including reproduction, are governed by the circadian clock [9]. Reproductive activity displays a regular cycle driven by complex interaction of the circadian system, hypothalamic neuropeptides, gonadotropins, and sex steroids produced by the ovaries [9]. Release of GnRH and GnRH neuron activity is known to display a circadian pattern [10]. These day/night differences could be generated by the endogenous clock located in the suprachiasmatic nucleus (SCN) [11]. The neuronal projections from the SCN to the preoptic area (POA) [12], where most cell bodies of GnRH neurons are located, suggest an influence of the SCN on the GnRH cell bodies in POA. Some studies on sheep have reported day/night differences in LH secretion. It was shown that, in ewes [13] and rams [14], a higher level of LH occurs at night. Misztal et al. [15], in an experiment on luteal-phase ewes, observed an increase in LH release during the first hours of darkness.

Data concerning differences in daily GnRH/LH secretion during different seasons in sheep are fragmentary. In our study, the hypothesis that there are differences in day/night secretion of GnRH/LH and in GnRHR activity in ewes in follicular phase of the estrous cycle and during anestrous season will be tested. Therefore, the aim of the undertaken studies was to determine the influence of the day time on GnRH gene and peptide expression in the POA and ME, LH gene expression in AP and LH plasma level, and GnRHR gene expression in ME and AP in anestrous and follicular phase ewes.

## 2. Materials and Methods

### 2.1. Animals and Experimental Design

The study was performed on Blackhead ewes (*n* = 24) during long day (LD) (8:16, June; *n* = 12) and short day (SD) (16:8, October; *n* = 12) periods. The animals were maintained indoors in individual pens and were exposed to natural daylight present at 52° N latitude and 21° E longitude. The ewes were in good condition, i.e., their body condition was estimated at a three according to a five-point scale in LD as well in SD group [16], and the animals were acclimatized to the experimental conditions for one month. To avoid isolation stress, animals always had visual contact with each other. The animals were fed a constant diet of commercial concentrates with hay and water available ad libitum, according to the recommendations proposed by the National Research Institute of Animal Production for adult ewes [17].

The experimental procedures were performed on ewes in two different reproductive statuses. In the LD photoperiod, the experiment was performed on the ewes (*n* = 12) in physiological seasonal anestrous, while in the SD photoperiod, ewes (*n* = 12) were taken in the follicular phase of the estrous cycle. To standardize the experimental condition, ewes from SD were synchronized by Chronogest^®^CR (Merck Animal Health, Boxmeer, the Netherlands) according to the method described in our previous studies [18,19].

The animals from each photoperiod were divided into two groups: day (*n* = 6) and night (*n* = 6). In the night experiments, all procedures on animals were performed in the darkness with the use of red light to avoid the stimulation of retina. Blood samples (8 mL) were collected in 15 min intervals for one hour before the euthanasia. After centrifugation in heparinised tubes, plasma was stored at −20 °C until assayed. The animals were euthanized 5 h after sunset (night experiment) or 5 h after the sunrise (day experiment). It is worth mentioning that the animals from the SD photoperiod were sacrificed in the second day of the follicular phase 29 h after PMSG injection in each group. The brain was immediately removed from the skull, and the anterior pituitary (AP), preoptic area (POA), and median eminence (ME) were dissected, immediately frozen in liquid nitrogen, and stored at −80 °C until further assay. All procedures were performed with agreement of the Local Ethics Committee of Warsaw University of Life Sciences (Warsaw, Poland; authorization no. 59/2011; 50/2013).

### 2.2. LH Assay

The LH concentration in plasma was determined by radioimmunoassay using anti-ovine-LH (and anti-rabbit-y-globulin antisera, IFZZ PAN Poland) and ovine standard (NIH-LH-S018) [20]. The sensitivity of assays was 0.3 ng/mL, and the intra-assay and inter-assay coefficients of variation were 8.3% and 12.5%, respectively. The medium LH level was expressed as the mean ± SEM of LH concentration in the media collected.

### 2.3. Melatonin Assay

Melatonin concentration in plasma was analysed by radioimmunoassay double-antibody method according to the method of Fraser et al. [21], and modified in our laboratory [22], using anti-ovine melatonin serum (AB/S/01, Stockgrand Ltd., Surrey, UK). Melatonin (Sigma-Aldrich, St. Louis, MO, USA) as a standard synthetic and (O-methyl-3H)-melatonin (AmershamPLC, Amersham, UK) as a tracer were used. The sensitivity of the assay was 16.8 ± 8.0 pg/mL and the intra- and inter-assay coefficients of variation were 10.5 and 13.2%, respectively.

### 2.4. GnRH Peptide Assay

The concentration of GnRH in the POA was determined with a commercial GnRH ELISA kit (CUSABIO BIOTECH Co., Ltd., Wuhan, China) dedicated for sheep samples. All steps in the assays were performed according to the manufacturer’s instructions. The tissues were homogenized in 400 µL of phosphate buffered saline (0.02 M). Then, homogenates were subjected to two freeze-thaw cycles to further break the cell membranes. After that, the homogenates were centrifugated for 15 min at 1500× *g* in 4 °C. The supernatants were aliquoted and stored until assay in −80 °C. The incubation of plates and absorbance measurement at 450 nm were performed using a MAX-MAT PL II reader (Maxmat S.A., Montpellier, France). The assay sensitivity was 0.5 pg/mL. The values of GnRH concentration were normalized to total protein content in each sample assayed using the Bradford method. The mean concentration of total protein per well in the GnRH peptide assay was 2.6 ± 0.2 mg/mL.

### 2.5. Real-Time PCR Analysis

The total RNA from the AP, POA, and ME was isolated using the NucleoSpin^®^RNA II Kit (MACHERRY-NAGEL Gmbh&Co.; Düren, Germany). The purity and concentration of the isolated RNA were quantified spectrophotometrically. The RNA integrity was confirmed by electrophoresis using 1% agarose gel stained with ethidium bromide. To synthesize cDNA, the Maxima™ First Strand cDNA Synthesis Kit for RT-qPCR (Thermo Fisher Scientific, Waltham, MA, USA) and 2 µg of total RNA were used.

Real-time PCR was performed using the HOT FIREPol EvaGreen^®^ qPCR Mix Plus (Solis BioDyne, Tartu, Estonia) and HPLC-grade oligonucleotide primers (Genomed, Warszawa, Poland). Specific primers for determining the expression of housekeeping genes and the genes of interest were designed using Primer 3 software (Table 1). One reaction mixture (total volume: 20 µL) contained 4 µL of PCR Master Mix (5×), 14 µL of RNase-free water, 1 µL of primers (0.5 µL each primer, working concentration 0.25 µM), and 1 µL of the cDNA template. The reactions were run on the Rotor-Gene 6000 instrument (Qiagen, Dusseldorf, Germany). The following protocol was used: 95 °C for 15 min and 30 cycles of 95 °C for 10 s for denaturation, 60 °C for 20 s for annealing, and 72 °C for 10 s for extension. A final melting curve analysis was performed to confirm the specificity of the amplification.

Relative gene expression was calculated using the comparative quantification option of Rotor Gene 6000 software 1.7 (Qiagen) based on the comparative threshold cycle method, which requires determining a fractional cycle number called the threshold cycle (Ct) (Rasmussen, 2001) [23]. Three housekeeping genes: GAPDH, ACTB, and PPIC, were tested. GAPDH was selected as the best endogenous housekeeping gene based on the result analysis performed with BestKeeper software. The results are presented in arbitrary units, as the ratio of the target gene expression to the expression of the housekeeping gene.

### 2.6. Statistical Analysis

The statistical analysis was performed using the STATISTICA 10 software (Stat Soft. Inc., Tulsa, OK, USA). The results of hormones and GnRH concentration as well as gene expression were analyzed using two-way analyses of variances (ANOVA) to identify the significant influence of two parameters (season and time of the day) and followed by a post-hoc Fisher’s test. The results are presented as the mean ± S.E.M. Statistical significance was set at *p* < 0.05.

## 3. Results

### 3.1. The Influence of the Day Time on the Secretion of LH in Anestrous and Follicular Phase Ewes

In the ewes during the follicular phase, the serum concentration of LH was higher (*p* < 0.05) than in anestrous animals, both in the day (7.2 ± 0.17 ng/mL vs. 4.3 ± 0.08 ng/mL) and at night (5.7 ± 0.33 ng/mL vs. 2.7 ± 0.14 ng/mL). It was also observed that the plasma level of LH was higher (*p* < 0.05) during the day compared with the nocturnal concentration of this hormone at both studied phases (Figure 1A). It was observed that the daily LHβ mRNA expression in the AP was higher (*p* < 0.05) in ewes in the follicular phase compared with anestrous animals. Expression of ewes in both the follicular as well as the anestrous phase LHβ gene was higher (*p* < 0.05) in the AP collected during the day compared with at night (Figure 1B).

### 3.2. The Effect of the Day Time on the Circulating Melatonin Concentration in Anestrous and Follicular Phase Ewes

The nocturnal concentration of melatonin in serum from ewes during the follicular phase (311.4 ± 40.01 pg/mL) was higher (*p* < 0.05) compared with the concentration in animals during the anestrous period (157.1 ± 10.46 pg/mL) (Figure 2).

### 3.3. The Influence of the Day Time on the GnRH Expression in the POA and ME in Anestrous and Follicular Phase Ewes

It was observed that the nocturnal GnRH content in the POA from ewes during the follicular phase (20.1 ± 2.87 pg/mg) was higher (*p* < 0.05) compared with the level of this peptide in ewes in the anestrous phase (7.3 ± 0.25 pg/mg). It was also observed that, in animals in the follicular phase, the concentration of GnRH at night (20.1 ± 2.87 pg/mg) was higher (*p* < 0.05) than during the day (10.8 ± 1.17 pg/mg) (Figure 3A). It was found that the nocturnal GnRH mRNA expression in the POA was higher (*p* < 0.05) in ewes during the anestrous period compared with animals in the follicular phase. Day/night differences in GnRH gene expression were observed in the POA where this gene expression at night was higher (*p* < 0.05) than in the day in the anestrous season and, inversely, in ewes during the follicular phase, GnRH mRNA was higher (*p* < 0.05) during the day (Figure 3B). In the ME, the daily GnRH mRNA expression was higher (*p* < 0.05) in anestrous ewes compared with animals in the follicular phase. It is worth mentioning that no mRNA encoding GnRH was found in the ME at night in animals during the anestrous season (Figure 4).

### 3.4. The Influence of the Day Time on the GnRHR Gene Expression in the ME and AP in Anestrous and Follicular Phase Ewes

It was found that the daily GnRHR gene expression in the ME is higher (*p* < 0.05) in follicular phase than in the anestrous season. On the other hand, the nocturnal level of GnRHR mRNA expression was lower (*p* < 0.05) in follicular phase compared with the anestrous season. Moreover, day/night differences in GnRHR gene expression were observed in the ME only in the follicular phase ewes, whereas the level of GnRHR mRNA was lower (*p* < 0.05) at night (Figure 5A). It was determined that the daily GnRHR gene expression in the AP is also higher (*p* < 0.05) in the follicular phase compared with the anestrous season. The nocturnal level of GnRHR mRNA expression in the AP was higher (*p* < 0.05) in the anestrous season, but lower (*p* < 0.05) in the follicular phase compared with the daily expression of this gene, respectively (Figure 5B).

## 4. Discussion

Our study is the first scientific report showing the day/night fluctuation of LH secretion in adult ewes regardless of the reproductive season. It was found that daily expression of LHβ mRNA in the AP as well as LH release is higher compared with the night in both the anestrous season and the follicular phase. It could be considered that the pattern of LH secretion in ewes enables adaptation for daily sexual activity of this species. It is likely that nocturnal suppression of LH secretion in the follicular phase may delay occurrence of the LH surge, shifting the ovulation closer to the day time. The study performed on the day-active rodent Arvicanthis niloticus showed that the LH surge occurs just before the active period, which was preceded by the accumulation of GnRH peptide in the hypothalamus [25]. Moreover, in another study, it was reported that, in women, the day/night rhythm of LH secretion also occurs [26]. Kerdelhue and coworkers [26] found that the preovulatory LH surge generally occurs between midnight and 08:00. The other study analyzing the timing of the LH surge in women reported that 48% of the surges occurred between 04:00 and 08:00 and 37% of the surges occurred between midnight and 04:00 [27].

Our study showed that the content of GnRH peptide in the POA characterized the day/night fluctuation, but only during the follicular phase. Unexpectedly, nocturnal expression of GnRH in the POA during the follicular phase was higher compared with the daily level of this peptide. This discrepancy with LH secretion may suggest that, during the night, GnRH is synthesized in the neuron perikarya, but it is not released into the portal blood system. On the other hand, it was observed that the gene expression of GnRH is reduced during the night in the follicular phase in the POA and hypothalamic structure, where the majority of GnRH neurons perikarya are located. However, no changes in the level of GnRH mRNA in the ME during the follicular phase were determined. In the anestrous season, no day/night changes in the level of GnRH peptide in the POA were found, but nocturnal stimulation of GnRH gene expression in this hypothalamic structure was determined. It is worth pointing out that no mRNA encoding GnRH was detected in the ME during the night in anestrous ewes. An interesting feature is that suppression of LH secretion after the nightfall seems to be constitutive regardless the reproductive status, when the mechanisms influencing the synthesis of GnRH differ between the anestrous and follicular phase.

The synthesis of GnRH in the POA may be controlled by the SCN, which controls a wide variety of circadian behavioral and physiological processes in mammals [28]. The studies on rodents revealed that a signal emanating from the SCN may influence the timing of the LH surge. In these animals, the lesions of the SCN interfered with ovulation process and even blocked the LH surge [29,30]. The SCN may influence the LH surge via projections onto hypothalamic GnRH neurons [30,31]. On the other hand, the SCN and GnRH system seems to be bidirectional, and GnRH input in the SCN region represents an anatomical substrate for feedback-control between these systems [32]. It was found that the SCN neurons that project to the POA contain vasoactive intestinal polypeptide (VIP), synthesized in cell bodies in the ventrolateral part of the SCN and vasopressin (VP), synthesized predominantly in the dorsomedial part of the SCN [33,34,35]. The other neural pathway of the SCN and GnRH neurons’ communication may be AVP neurons [36]. The study on monkeys showed the existence of synapses between AVP terminals and GnRH neurons in supraoptic nucleus [36]. However, the results of the histological studies on sheep are ambiguous. In sheep, few AVP terminals were found in the POA close to GnRH neurons without direct contact between them. Moreover, GnRH cell bodies in the anterior supraoptic nucleus, with dense AVP neuron innervation, were not in direct contact with AVP terminals [37].

However, there is no strong evidence for direct action of the AVP/VIP neurons in the modulation of GnRH neurons activity in sheep. It could be speculated that AVP neurons regulate GnRH secretion indirectly via influencing the activity of other neural pathways [37].

The photoperiod-dependent changes in the activity of GnRH neurons may not exclusively result from the modulatory role of the SCN, but could be influenced by other factors showing day/night fluctuation. It is known that the secretory activity of GnRH neurons is regulated by multiple neural systems that involve many neurotransmitters, neurohormones, and peptides; some of them could be secreted in daytime-dependent mode [4]. The studies on rats showed daytime-dependent variations in the production of catecholamines in the MPOA [38]. The differences were demonstrated in the concentration of norepinephrine (NE) and dopamine (DA) between midday and midnight levels of these amines. Dopamine and their metabolite DOPAC concentrations were higher during the day, while the noradrenaline level was higher at night [38]. The differences in the pattern of day/night GnRH peptide content in the anestrous season and follicular phase could result from the fact that the role of catecholamines in the regulation of GnRH/LH secretion varies between seasons [39]. The results of Scott et al. [39] provide evidence of the involvement of the noradrenergic system in the regulation of GnRH secretion at the level of the GnRH cell bodies in the preoptic area, with clear influences of season and E2 status on this regulation. In anestrous ewes, when the E2 level is low, NE inhibits GnRH/LH secretion, while in estrous ewes, before the preovulatory surge, NE stimulates GnRH release [40]. As monoamines play a crucial role in the regulation of GnRH release in ewes, the day/night fluctuations in the content of these amines could influence the day/night differences in the GnRH/LH secretion observed in our experiment.

Moreover, in the study on rats and hamsters, the day/night differences in the monoamine contents were also observed in the pineal gland, another important component of the circadian system under the SCN influence [41]. It is well known that melatonin is synthetized in the pineal gland and that this hormone reflects the environmental light conditions [42]. This gland is localized in a third ventricle in place called the pineal recess [43]. The local concentration of melatonin in the third ventricle is approximately 20 times higher in the CSF than in blood [44]. Melatonin synthesis and release is regulated by norepinephrine, released from sympathetic nerve fibers exclusively at night [41]. The study on sheep showed that the nocturnal concentration of melatonin as well as duration of its release depend on photoperiodic conditions [22,45]. Our present study confirmed that the highest level and longest period of melatonin secretion occurs in the winter during the short day photoperiod. The role of melatonin in the regulation of HPG axis in sheep is still elusive. The LH secretion may be affected by melatonin at the level of the pituitary gland. It was previously showed that a potential target for melatonin action could be gonadotropes placed in the pars tuberalis (PT), which contains the highest density of melatonin receptors among other adenohypophyseal subdivisions [46]. In our previous ex vivo studies, we found that basal and GnRH-stimulated secretion of LH in the PT is higher than in the pituitary [47,48]. The results of our in vivo study showing reduction of LH secretion five hours after nightfall are consistent with these ex vivo experiments [48], and suggest that melatonin acting at the level of the pituitary may be involved in the reduction in circulating LH concentration. PT contains the gonadotropic cells and secretes the LH and FSH [49,50]. The results from the studies of Lafarque et al., [51] showed that the activity of the PD may be regulated and supported by the secretory activity of the PT.

However, melatonin may also affect the activity of the HPG axis at the hypothalamic level. The study performed by Malpaux et al. [52] on ovariectomized ewes with micro-implants containing melatonin placed in different hypothalamic areas including the POA, anterior hypothalamus, dorsolateral hypothalamus, and MBH showed that increased release of LH was found only in animals with the implants placed in the MBH. It is worth noting that melatonin receptors were either not found in the sheep hypothalamus or were detected in small numbers in places that are not involved in reproduction [15]. In the POA, the structure where the most of GnRH-neurons pericarya are localized melatonin failed to stimulate the GnRH/LH secretion [53,54]. Finally, Malpaux et al. [55] showed the occurrence of melatonin binding sites in the premammillary hypothalamic area and that melatonin stimulates LH secretion if it is delivered into this site. This suggests the premammillary hypothalamus is an important target for melatonin in regulating reproductive activity in ewes. However, the influence of the melatonin on GnRH/LH secretion at the hypothalamic level may be influenced by the reproductive status. Romanowicz et al. [56] showed that intracerebroventricular (icv.) infusion of melatonin did not affect both GnRH and LH secretion in anestrous ewes. On the other hand, the same research group found that intracerebroventricular (icv). administration of melatonin stimulated LH secretion in ovariectomized ewes during the breeding season [57]. This seems to support our present observation that the nocturnal content of GnRH peptide in the POA was not changed during the anestrous season, but was significantly elevated in the follicular phase.

Another factor influencing the HPG axis, whose secretion shows day/night fluctuations, is interleukin (IL)-1β [22]. Our previous studies showed that, acting at the level of hypothalamus, proinflammatory cytokines, particularly IL-1β, are potent negative regulators of GnRH secretion in ewes [58,59,60]. The study on rats showed that there are day/night changes in the hypothalamic IL-1β mRNA expression. IL-1β mRNA was the highest after the lights were turned on, declined during the remaining light period, and was lowest in the dark period [61]. This suggests that observed changes in the GnRH secretion did not result from the changes in this cytokine action because our studies were performed on ewes taken 5 h after the sunrise and 5 h after the sunset, when the brain expression of IL-1β in healthy animals is low.

Our study also suggests that the sensitivity of the AP cells to the GnRH stimulation may be dependent upon the photoperiodic condition. It was found that, in anestrous ewes, the night expression of GnRHR mRNA was higher, but during the follicular phase, night expression of GnRHR gene was reduced compared with the daily gene expression of this receptor. This is not surprising considering that, similarly to GnRH gene expression and its regulation, the GnRHR mRNA level is different during the anestrous and breeding season. It was shown that GnRHR gene expression in anestrous ewes is lower than in the follicular and/or luteal phase ewes [62]. It must be pointed out that GnRHR gene expression is regulated by a complex system of interactions, including E2, P4, and GnRH, which all, as has already been mentioned above, show seasonal and circadian changes [4]. It could be speculated that the differences in diurnal and seasonal changes in the GnRHR gene expression observed in our experiments may result from the fact that these factors exert distingue effects in the estrous cycle and anestrous season; however, this issue requires future detailed studies.

## 5. Conclusions

Summarizing, our study showed that GnRH/LH secretion is subject to diurnal and seasonal changes. There are multiple possible mechanisms influencing circadian changes in the GnRH/LH secretion; however, the reduction of LH release a few hours after the sunset seems to be universal for both the anestrous season and follicular phase; the processes occurring at the hypothalamic level are more equivocal. Owing to the fact that a sheep is a day-active animal, it could be concluded that the suppression of LH secretion in the follicular phase at the beginning of the night may be a mechanism moving the LH surge to the early morning.

## Figures and Tables

**Figure 1 animals-12-00283-f001:**
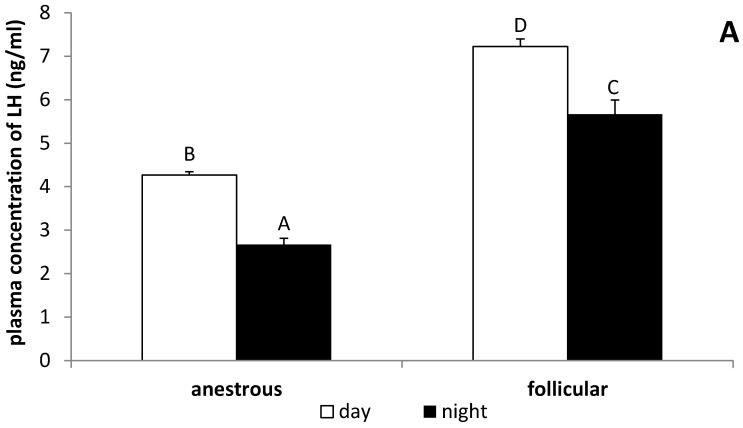
The day/night changes in the circulating level of luteinizing hormone (LH) (**A**) and gene expression of LHβ (**B**) in the anterior pituitary (AP) in the anestrous and follicular phase ewes. Gene expression data were normalised to the average relative level of LHβ gene expression measured in the anestrous animals from the day experiment, which was set to 1.0. All data are presented as the mean (±S.E.M.). Different capital letters indicate significant (*p* < 0.05) differences according to a two-way ANOVA followed by NIR Fisher’s post-hoc test.

**Figure 2 animals-12-00283-f002:**
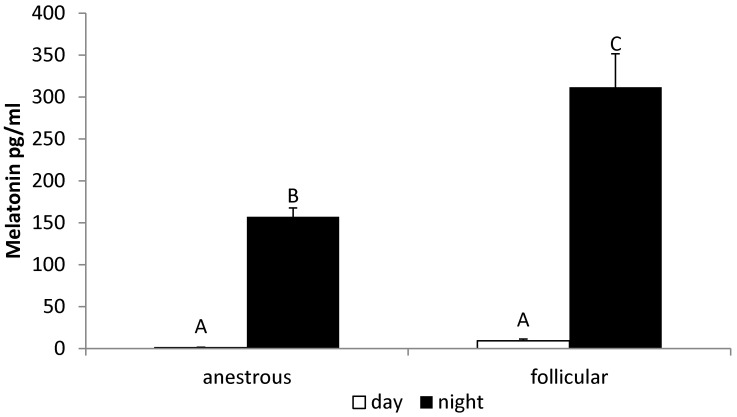
The day/night changes in the circulating level of melatonin in the anestrous and follicular phase ewes. All data are presented as the mean (±S.E.M.). Different capital letters indicate significant (*p* < 0.05) differences according to a two-way ANOVA followed by NIR Fisher’s post-hoc test.

**Figure 3 animals-12-00283-f003:**
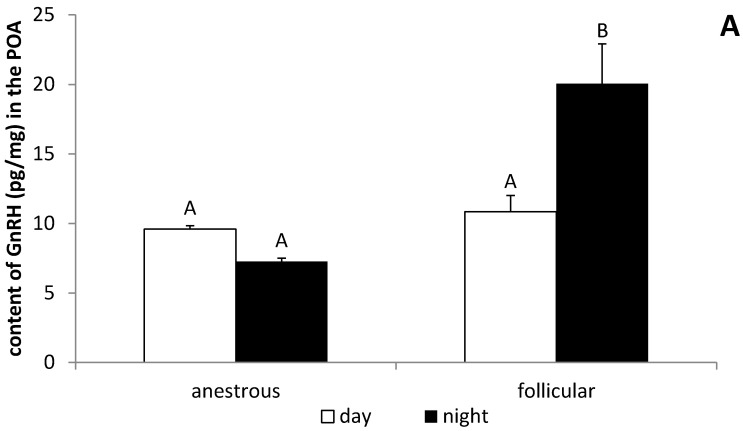
The day/night changes in the GnRH peptide (**A**) and mRNA (**B**) expression in the preoptic area (POA) of the anestrous and follicular phase ewes. Gene expression data were normalised to the average relative level of GnRH gene expression measured in the anestrous animals from the day experiment, which was set to 1.0. All data are presented as the mean (±S.E.M.). Different capital letters indicate significant (*p* < 0.05) differences according to a two-way ANOVA followed by NIR Fisher’s post-hoc test.

**Figure 4 animals-12-00283-f004:**
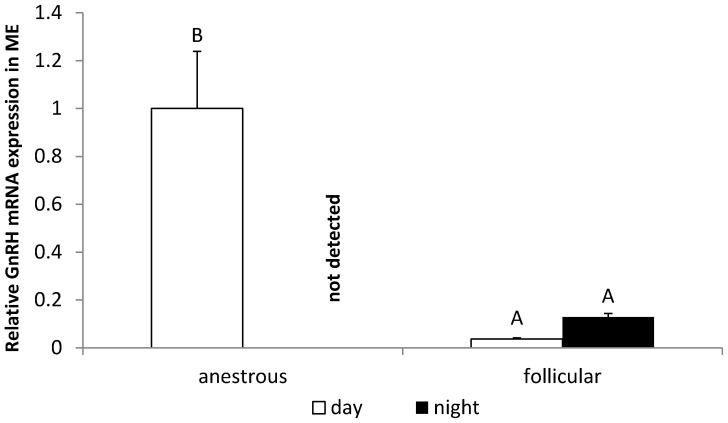
The day/night changes in the relative gene expression of GnRH in the mediane eminence (ME) of the anestrous and follicular phase ewes. Gene expression data were normalised to the average relative level of GnRH gene expression measured in the anestrous animals from the day experiment, which was set to 1.0. All data are presented as the mean (±S.E.M.). Different capital letters indicate significant (*p* < 0.05) differences according to a two-way ANOVA followed by NIR Fisher’s post-hoc test.

**Figure 5 animals-12-00283-f005:**
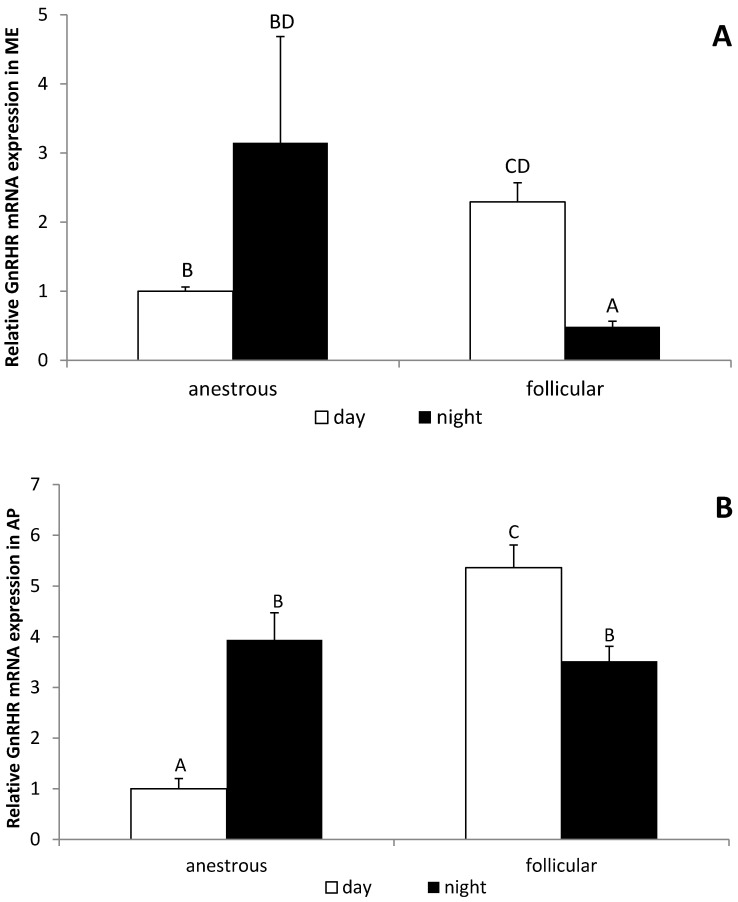
The day/night changes in the relative gene expression of gonadotropin releasing hormone receptor (GnRHR) gene expression in the median eminence (ME) (**A**) and in the anterior pituitary (AP) (**B**) of the anestrous and follicular phase ewes. All data are presented as the mean (±S.E.M.). Different capital letters indicate significant (*p* < 0.05) differences according to a two-way ANOVA followed by NIR Fisher’s post-hoc test.

**Table 1 animals-12-00283-t001:** All genes analyzed by real-time PCR are listed with their full names and abbreviations.

GenBank Acc. No.	Gene	Amplicon Size (bp)	Forward/Reverse	Sequence 5′ → 3′	Reference
NM_001034034	*GAPDH* glyceraldehyde-3-phosphate dehydrogenase	134	forward	AGAAGGCTGGGGCTCACT	Herman et al. [24]
			reverse	GGCATTGCTGACAATCTTGA	
U39357	*ACTB* beta actin	168	forward	CTTCCTTCCTGGGCATGG	Herman et al. [24]
			reverse	GGGCAGTGATCTCTTTCTGC	
NM_001076910	*PPIC* cyclophilin C	131	forward	ACGGCCAAGGTCTTCTTTG	Herman et al. [24]
			reverse	TATCCTTTCTCTCCCGTTGC	
X52488	*LHβ* luteinizing hormone beta-subunit	184	forward	AGATGCTCCAGGGACTGCT	Herman et al. [24]
			reverse	TGCTTCATGCTGAGGCAGTA	
NM-001009397	*GnRHR* gonadotropin-releasing hormone receptor	150	forward	TCTTTGCTGGACCACAGTTAT	Herman et al. [24]
			reverse	GGCAGCTGAAGGTGAAAAAG	
U02517	*GnRH* gonadotropin-releasing hormone	123	forward	GCCCTGGAGGAAAGAGAAAT	Herman et al. [24]
			reverse	GAGGAGAATGGGACTGGTGA	

## Data Availability

The data presented in this study are available on request from corresponding authors.

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
