# Peer review of "The Effect of Photoperiodic Conditions on GnRH/LH Secretion in Ewes"

_animals, 2022, doi:10.3390/ani12030283_

Round 1

Reviewer 1 Report

Dear Ms. Charlotte Shen
Assistant Editor of Animals

The manuscript is adequately developed even if some parts need to be modified to be more consistent with the study.
The purpose is not very incisive it should be more centered and describe precisely what you want to do.
Different season change in two season.
The methods are complete and adequately describe all the methods used.
The results show the data that have been analyzed even if a method should be found to bring together the figures more, otherwise the reading becomes dispersive.
The discussion is too long, the AA should focus more on discussing the results of the work and not get lost in considerations that are not appropriate for the study.
In my humble opinion the manuscript can be accepted after a minor revision.

Kind regards

Vincenzo Carcangiu

Author Response

Dear Sir,

thank You very much for Your comments. I tried to improove manuscript according to Your suggestions. I add more precise aim of the study at the end of Introduction (in blue colour). I tried to delete few parts of disscussion to focus more on discussing the results.

Sincerely 

prof. D. Tomaszewska-Zaremba

Reviewer 2 Report

I believe the manuscript has been significantly improved and
now warrants publication in Animals. However, still  please indicate the concentration of total protein per well in the GnRH peptide assay.

Author Response

Dear Sir,

thank You very much for Your comments. I add the concentration of total protein per well in GnRH assay in Material and Metrhods in part concerning GnRH assay.

Sincerely 

prof. D. Tomaszewska-Zaremba

This manuscript is a resubmission of an earlier submission. The following is a list of the peer review reports and author responses from that submission.

Round 1

Reviewer 1 Report

The manuscript reports on the effect of photoperiodic conditions on GnRH / LH secretion in ewes. The study is interesting as it reports the different secretion and expression of the different hormones involved in regulating seasonal reproductive activity in sheep. It is now recognized that the regulation of the seasonal secretion of GnRH and LH is a rather complex mechanism and is linked to several factors that are not yet fully identified, much less the influence that these may have in the control of the secretion of these hormones. I will not enter into the goodness of the data obtained and the importance of these results they may have to understand the regulation of GnRH secretion but on the disquisitions made in the discussion. The data in the possession of the authors do not allow to argue about the various factors that can influence the hypothalamic secretion of GnRH as they have no basis because they have not been analyzed. The only data that could be used to understand the seasonality of GnRH and LH secretion is the melatonin levels that were not used. Considering that the authors possess the RNA and serum of these samples they could add data and substantiate their hypotheses with reliable data.

Surely the addition on the expression of other factors at the level of the CNS areas analyzed will lead to important discoveries that can support all the assumptions made in the manuscript.

Therefore in my humble opinion the study must be supplemented with other data in order to be published in this journal.

Reviewer 2 Report

General comments

This study examines differences in day/night LH and melatonin concentrations, GnRH/LH expression in ewes in follicular phase of the estrous cycle and during anestrous season. The aim of the study may surely be of interest, however there are several flaws which significantly decrease the quality of the paper.

The main concern is the experimental design. As stated in the M&M section (lines 103-106), in long-day photoperiod (June) the experiment was performed on the ewes in physiological seasonal anestrous, when in short-day photoperiod (October), ewes were in the follicular phase of the estrous cycle. Under these conditions, Authors cannot distinguish between the effect of the photoperiod and that of reproductive activity on the variable analyzed. In addition, to synchronize the follicular phase, ewes from the SD group were administered a dose of PMSG. This treatment alters GnRH/LH secretion. Thus, this factor adds to the confounding effects of photoperiod and reproductive cyclicity.  Therefore, the conclusion and interpretation of the results is questionable.  

Specific comments

Lines 32-38: please re-write, sentence too long and difficult to follow in place;

Lines 84-88: specify the underlining hypothesis and the impacts of the study;

Line 96: please express BCS as a mean specific for each group;

Line 116: at what hour of the day was PMSG administered?

Lines 136-141: was GnRH dosed in the cell extract? Better describe please.

Lines 173-178: incomplete description of the statistical analyses. How were differences in all other variables assessed?

Figure 1A: are those mean values from the 4 time points? Why not kept separated to see the pulse?

Discussion way too long and difficult to follow in place.

Reviewer 3 Report

The manuscript is very interesting, informative and complete in its experimental design. However, my concern is regarding the lack of circulating estradiol measurement, as well as determination of the expression of estrogen receptors in the selected parts of the ovine brain which were studied. For sure this aspects would raised the scientific value of the paper. Is the any reason why the Authors did not take estradiol in to account?

Other minor comments are:

- In the Abstract section only the full definition of LH is given and the full definition of GnRHR, GnRH and LHβ are not given.

- In the Material and methods section please describe the protein isolation method used for the GnRH peptide assay. Also please indicate the concentration of total protein per well in the GnRH peptide assay.

- In the Real-Time PCR analysis please describe in more detail the „ the comparative quantification option” which was used.

- Please clarify why LHβ expression was determined using only mRNA level not also protein level.

- In figure 4 the Authors maybe consider to use the wording „non detected” which is non determinative instead of „the lack of” which is determinative.

- Furthermore the total „Discussion” is too lengthy and the Authors should consider shortening it by reducing or removing the points around VIP and Kp,  which are not directly related to the experiments results.